# One City for All? The Characteristics of Residential Displacement in Southwest Washington, DC

**Flora Lindsay-Herrera**

Department of Library and Information Science, Catholic University of America; Washington, DC 20064, USA; lindsayherrera@cua.edu

**Abstract:** This paper examines two periods of renewal in Washington, DC, USA's southwest quadrant and their relationship with displacement. The paper situates this discussion within both the local historical continuum and globally-recognized paradigms, such as "the right to the city". This article primarily serves as an overview of urban planning consequences in Southwest Washington DC based on extant academic literature and policy briefs. Compared with the abrupt physical displacement in the 1950s and 1960s precipitated by a large-scale federally funded urban raze and rebuild project, urban planning in present-day DC includes mechanisms for public engagement and provisions for housing security. However, countervailing economic incentives and rapid demographic changes have introduced anxieties about involuntary mobility that the literature suggests may be born out of forced or responsive displacement. Two potential case studies in the area warrant future study to understand present-day mobilities in the context of the economic and socio-cultural factors shaping the actions of present and prospective residents and decision-makers.

**Keywords:** displacement; relocation; urban planning; neighborhood change

---

## 1. Introduction

Following World War II, the southwest (SW) quadrant of the United States capital city, Washington, DC, was the site of a federal government-led urban renewal process that set a legal precedent for eminent domain, which was one of the largest acquisitions of land by the United States government to date, and pioneered new housing practices. The plan displaced 1500 businesses and 23,000 residents from 560 acres of land [1]. Despite the architectural significance of the site and the plan's international renown at the time, the SW urban renewal area has popularly been characterized as a failed attempt at downtown revitalization [2,3].

Washington, DC entered a boom of urban growth in the early 2000s, coupled with increasing income inequality and a severe affordable housing crisis. The SW quadrant has again become the focus of renewal initiatives, including the Wharf public-private partnership and the planned redevelopment of the nearby Greenleaf Gardens public housing project [4].

Literature on urban planning policy and the demographics of displacement in Washington, DC tend to highlight neighborhoods in the city's northwest quadrant as a case study [5–7]. This paper turns the focus on the lesser-studied SW quadrant and situates current public debate over renewal in the SW and its relationship to displacement within both the local historical continuum and globally-recognized paradigms, such as "the right to the city". The purpose of this paper is to highlight prospective avenues of study to understand if and how approaches to inclusivity in planning policy and planning practices have substantively changed, and how principles of equity and inclusivity are being enacted by different stakeholders in the Global North.

One must analyze the nexus of gentrification and displacement when examining mobilities in present-day Washington, as well as the role of governance vis-à-vis urban planning in these two

phenomena. The Encyclopedia of Housing [8] defines gentrification as "the process by which central urban neighborhoods that have undergone disinvestments and economic decline experience a reversal, reinvestment, and the in-migration of a relatively well-off, middle- and upper middle-class population" (p. 469). Zuk et al. [9] cautioned that "drawing an analytical distinction between gentrification and displacement is critical" (p. 41). Empirical evidence by Freeman [8] linking gentrification to displacement is insufficient; demographic change may be ascribed to the characteristics of in-movers. Research by Chapple [10] showed that rent appreciation and high-rent burden (relative to income) predict displacement, but that gentrification does not. Freeman [11] concluded that public housing residents who are able to stay in gentrifying neighborhoods because the rent is subsidized are more likely to be employed and have modestly higher earnings than their counterparts living in public housing where the surrounding neighborhoods are not gentrifying. The role of public investment in spurring residential displacement is also inconclusive [9].

Sturtevant's [12] study of the socioeconomic characteristics of in-movers and out-movers in DC concluded that less educated households are more likely to move both within the city and to the suburbs (relative to not moving) for both white and black households, suggesting evidence of displacement. Sturtevant added that "in most, but not all, examples of gentrification in U.S. cities, the new residents are middle- or upper-class whites while those being displaced are lower income African Americans" (p. 277). Although this contradicts the findings of McKinnish et al. [13] who suggested that "gentrification of predominantly black neighborhoods creates neighborhoods that are attractive to middle-class black households" (p. 180), the latter study examined census data from the 1990s, whereas Sturtevant studied DC between 2000 and 2010, a time with a marked influx of white residents.

In addition to economic considerations, studies weight social factors that may precipitate urban migration in U.S. cities. As observed by Sturtevant [12], Green et al. [5], and Hyra [7] in their studies of gentrification and displacement in Washington, DC, cultural dissonance and preferences may play a role in mobility as well. Hyra [7] examined the connection between cultural displacement and concomitant political displacement at the neighborhood level, specifically for long-term African-American residents who, for decades, formed the bedrock of many Washington, DC communities. Farrar [14] asserted that "the same discourses which help constitute urban space also inform who can claim legitimate access to that space" (p. 128). In a city where it is commonplace in white, transitory social circles to hear the incorrect assertion "no one is from DC", this article encourages scholars to consider the ongoing negotiations for access to the SW part of the city through several high-profile redevelopment projects as fertile grounds for in-depth community-level study [15].

## 2. Materials and Methods

This paper is an elaboration of a presentation in the panel session "Land Governance in the Global North: Pointing the Lens at the Developed World" at the Netherlands Academy on Land Governance's 2018 conference "Land governance and (Im)mobility: Exploring the nexus between land acquisition, displacement and migration" in Utrecht, the Netherlands. The intent was to introduce SW Washington, DC's historical experience with urban planning and mobility to a broader array of practitioners and provide possible case studies that may be of interest to researchers for future, more rigorous, sociological, economic, or anthropological analysis of urban planning and mobility at the community level.

This article primarily serves as an overview of urban planning decisions in SW Washington DC based on extant academic literature and policy briefs. It also draws on historical and primary source texts generated by policy-makers, community media sources, and other social actors involved in urban planning efforts in SW Washington to illustrate the trends observed in the literature, and to suggest areas for more in-depth sociological or demographic study. Findings summarized here focus primarily on the impacts on residents and residential housing, and not the related, but distinct, effects on locally-owned commercial enterprises.

## 3. Results and Discussion

### *3.1. Midcentury Southwest: A Showcase for Urban Renewal*

#### 3.1.1. Rationale for Renewal

The 1950 Comprehensive Plan for Washington, DC developed by the National Capital Planning Commission suggested the SW as possible urban redevelopment site. Motives for urban renewal cited by planners included: to reduce blight, to increase the city's tax base, to reverse suburban out-migration, and to improve physical quality of the area near the U.S. Capitol building [1,16]. While conducting a survey of the Southwest quadrant in 1942 as a potential site of wartime housing, Arthur Goodwillie, director of the Home Owners Loan Corporation, observed that the area had brick housing interspersed with older frame houses: "The latter are in a lamentable state of repair, dangerous, unhealthful, vermin- and rat-infested. They constitute a serious fire, safety, and health hazard and should be demolished, as a slum clearance measure, at an early date" [1] (p. 15).

The Southwest Urban Renewal Area defined in the early 1950s encompassed 560 acres to the south of the National Mall, a 3-km-long lateral axis of the city center anchored by the U.S. Capitol building on the east and the Lincoln Memorial to the west (Figure 1). Authority for the redevelopment project stemmed from a series of acts passed by Congress, as DC was under direct rule of the federal government from the 1870s to the 1970s. These acts enabled the Redevelopment Land Agency to plan and rebuild blighted areas, and provided authorized federal funds for redevelopment as well as relocation costs for residents and businesses [1].

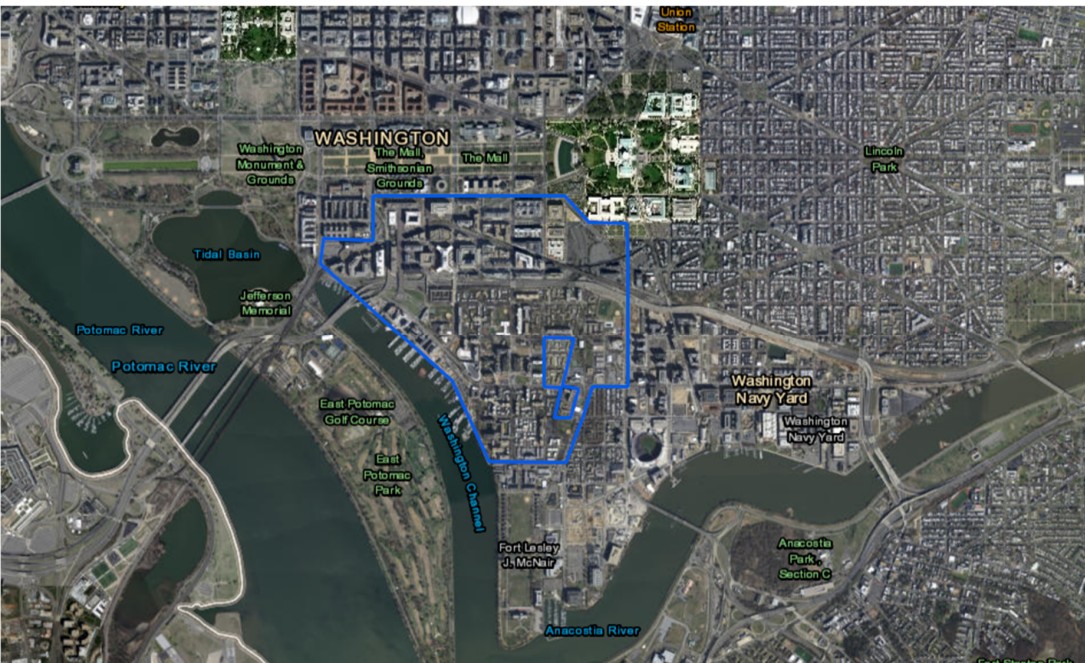

**Figure 1.** Southwest (SW) urban renewal area bounded area in blue. Boundaries of SW Urban Renewal Area adapted from Longstreth [17]. Basemap source: [18].

Even Goodwillie's assessment noted that not all buildings in the renewal area were derelict, and the quadrant had bustling commercial thoroughfares. The owners of a department store and hardware store in SW brought a case to the Supreme Court challenging the taking of unblighted properties on the grounds that portions of the 1945 Redevelopment Act were unconstitutional under the takings clause of the United States' Constitution's 5th Amendment. However, the Supreme Court ruled that the government can transfer property from one private party to another as part of a redevelopment plan that serves a public purpose, provided there was just compensation to the property owner [19,20].

3.1.2. Characteristics of the Renewal Zone

From 1954 to 1973, the Redevelopment Land Agency (RLA) acquired land by negotiated purchases or eminent domain in three designated sub areas (Area B, Area C-1, and Area C). The RLA then sold or leased land to project developers. The RLA razed 99% of the project area, demolishing 4800 structures. Of the estimated population in the urban renewal zone in 1950, nearly 90% were classified as low income and 69% African-American; 81% of the units were tenant-occupied [21]. In contrast, African-Americans constituted 35% of Washington residents in the city according to the 1950 U.S. census [22]. The redevelopment displaced 1500 businesses and 23,500 people from 560 acres [1]. Of the businesses, 59% were relocated, 15% remained, and 26% went out of business [1].

The Health and Welfare Council of DC conducted a demonstration program of community organization and social services for residents in Area C. The program included counseling by experienced case workers, group education, and coordinated social services, including public health nurses [23]. A population sample taken in 1960 of the 198 demonstration families found that [23]: 87% rented their units; 65% of sample residents had been in SW more than 10 years; 69% of the sample population of 23,416 were black (80% of demonstration families); 88% of the sample population was categorized as low income, 12% were moderate income; 76% of dwellings were substandard; and of the demonstration families, 43% had outside toilets, 70% had no central heating, and 21% had no electricity.

3.1.3. Short-Term Impacts of the Urban Renewal Project

The new SW has been historicized as marking racial inclusivity but economic segregation. Daniel Thursz, the sociologist tasked with conducting follow-up to the Housing and Welfare Council demonstration project, observed in 1966 that public opinion of the renewal plan at the time was paradoxical, with "myths and conflicting allegations": "Still others see urban renewal as 'Negro removal', while some fear a process which seems to bring racial integration to once racially homogenous areas" [23] (p. 5). For example, the Capitol Park Apartments opened in 1959, and by 1963 "about 10% of the apartments and between 20 and 25% of the houses were occupied by blacks" [18] (p. 276). The project introduced condominium ownership at the Tiber Island complex, one of the first such arrangements in the metropolitan area and an early example of its kind nationally [18] (p. 271). Nearly 60% of residents with known prior addresses came from outside of DC [21].

However, the RLA removed the low rent requirements for the new complexes during negotiations with developers, as a result of cost increases and lack of legislation, to make lower-middle income housing economical as well as public housing elsewhere in SW [1]. The scaling back of provisions for lower income residents of the area to remain in the zone led to the dispersion of many in the SW community to other quadrants of the city. Some families (at least 200 in Area B) moved on their own [17], although Thursz noted that, in the Area C zone of 17,500 persons, "the more able, independent, and well-to-do families moved without help from officialdom" resulting in a shift toward lower-medium income households remaining in the zone by the time plots were cleared [23] (p. 3). For residents, the RLA provided relocation assistance with the guarantee that: "You are assured the opportunity of being rehoused in an accommodation that is decent, safe, and sanitary, at a rent or price within your means, in an area reasonably accessible to places of employment and not generally less desirable in regard to public utilities and public commercial facilities than the area in which you now live" [23] (p. 145).

Of those residents tracked in the Health and Welfare Council of DC's demonstration project, they relocated to 37 census tracts throughout the city in a "shotgun" pattern [23] (p. 99). Displaced families were given priority for the Channel Square apartments built in 1966 and the rehabilitated philanthropic housing complex of St. James Mutual Homes both within the zone [17]. Some from Project Area B moved into Arthur Capper Dwellings, Kenilworth Courts, and Greenleaf Gardens, which were being built at the time, and the last of which was located immediately adjacent to the renewal zone (Figure 2). However, those who moved in later phases of the relocation faced limited housing alternatives [23].

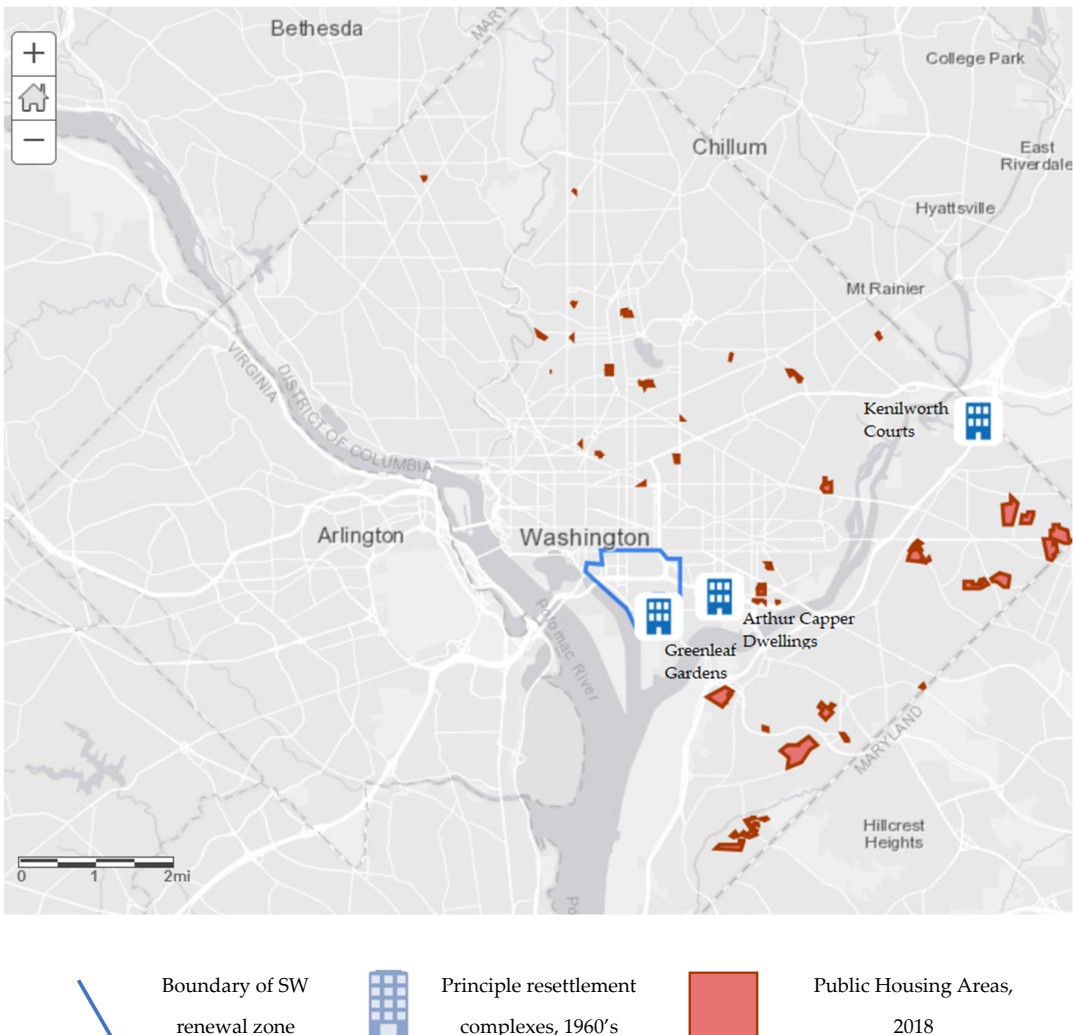

Figure 2. Overlay of (left to right) Greenleaf Gardens, Arthur Capper Dwellings, and Kenilworth Courts with additional present-day public housing areas, predominantly in Washington's east and south quadrants. Source of 2018 public housing areas: [24].

When Thursz conducted follow-up visits in 1996, participants noted an overall increased quality of housing, with 85.7% in dwellings rated as "good" [23] (p. 28); participants also noted better garbage collection and street cleaning services [23]. However, Thursz observed a decreased social connectedness among relocated families that "increases the destruction of social fabric and makes it difficult, if not impossible, for the former residents to maintain an ongoing sense of community with their former neighbors" [23] (p. 100) including the (i) drop in use of community resources and manifestations of nostalgia for organizations and settlement house of the old Southwest and (ii) the fact that only 4.2% respondents indicated they maintain frequent and reciprocal contact with former neighbors. Of the respondents, 42.1% had made three or more new friends but 26.3% of respondents had not made any new friends [23].

Thursz closed his study with recommendations for future projects, the tone of which is echoed in present-day planning documents for the quadrant. Recommendations from the Thursz study included the retention of social fabric through relocation to homes on or near the project site, and emphasis on funds and provision of neighborhood services, as well as the need for additional research on displacement [23].

### 3.2. Then to Now—Changing Paradigm for Urban Planning

In the 1970s across the United States, the approach to urban renewal shifted from large-scale to more selective rehabilitation and spot clearance. Simultaneously, the District of Columbia Home Rule Act of 1973 established a popularly elected mayor and city council, as well as elected advisory neighborhood commission (ANC) representatives. Establishment of the ANC system created avenues for community input. A new DC Comprehensive Plan was prepared under home rule, with neighborhood plans to bring zoning into conformance with the Comprehensive Plan [25].

Against this backdrop, the reputation of the SW urban renewal drew more critique. One urban studies assessment of Washington, DC in the 1980s characterized the renewal zone as one of the "worst excesses of the federal Urban renewal program" [25] (p. XI), vividly portraying: "A swath of suburbia infected with shopping mallaria . . . .cheek by jowl with the affluence of the new southwest is the Greenleaf public housing project . . . These sharp race and class distinctions have created tensions, which persist to the present day"(p. 58). Another report submitted to the United States Commission on Civil Rights described it as "notorious as a demonstration of the conviction that wholesale rebuilding was a benefit outweighing the social cost of wholesale displacement" [26] (p. 13). A 2006 National Capital Planning Commission report was ambiguous on its perceived success, simply noting that "The middle-class families who were to have occupied the housing did not return" [16] (p. 273).

A 2018 exhibit at the Anacostia Community Museum—a federally funded museum under the Smithsonian Institution—highlighted SW and other DC neighborhoods in 2018 with an exhibit on a history on neighborhood change and civic engagement. In a media interview, the exhibit's curator stated, "Southwest was ground zero in many ways" [27]. The exhibit, which shifts attention from the technocratic assessment of planning processes back to the perspective—and agency—held by the city's former and present residents, situates the urban change in SW squarely within the current of global dialogue on urban sustainability through its title: "A Right to the City".

### 3.3. Forces of Displacement in Present-Day DC

#### 3.3.1. Systemic Inequalities Reflected in Settlement Patterns

Historical migration patterns and exclusionary housing policies, such as red-lining, have a visible impact on population distribution in DC today (Figure 3). As private and public investment moves east, neighborhoods that were primarily lower income and African American are experiencing new and infill development, rising housing prices, and an influx of new residents. The stark visual breakdown of income distribution, racial distribution, and gentrification in DC have compelled calls for equity in new planning initiatives.

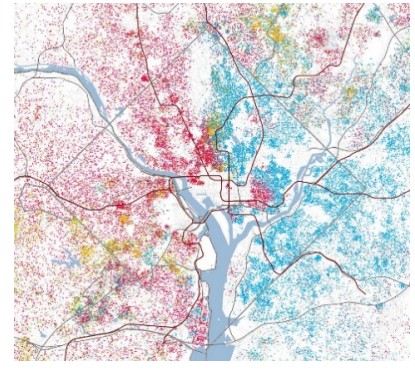

Key: ● = white; ● = black; ● = Asian; ● = Hispanic

**(a)**

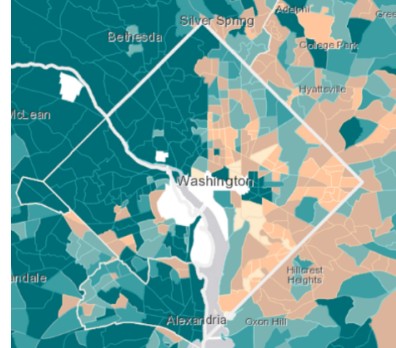

Key: ● USD $24,000 or less; ● $24,001–$39,000; ● $39,001–$53,000; ● $53,001–$68,000; ● $68,001–$82,000 ● More than $82,000

**(b)**

**Figure 3.** *Cont.*

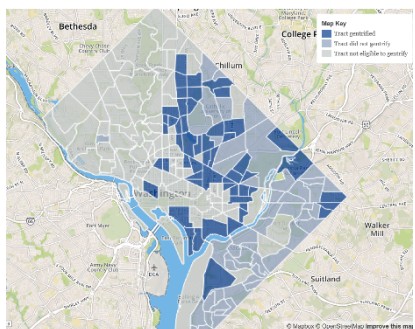

Key: ● Tract gentrified; ● Tract did not gentrify; ● Tract not eligible to gentrify

**(c)**

**Figure 3.** (**a**) Population distribution in Washington, DC by race. Each dot represents 25 people. Source: [28]; (**b**) Median household income. Source: [29]; (**c**) Gentrification trends: 2000–2013 using census (2000) and American Community Survey data (2009–2013) Source: [30].

3.3.2. Potential Drivers of Forced Displacement in Washington, DC

Zuk et. al. categorized the causes of displacement into forced displacement, including through the federal Housing Choice Voucher Program (Section 8), discrimination or condo conversion, and responsive displacement such as rent increases and increased taxes. Other factors in responsive displacement cited by Zuk et al. [9] included cultural dissonance and lack of social networks. Drivers in DC that may lead to forced displacement include:

1. Rapid population growth. The city's population growth of 13.2% from 2010 to 2016 has outpaced a 5% increase in housing stock [31].
2. Housing size preferences. Preference by smaller households to occupy larger units (both rent and purchase) resulting in increased competition for family-sized units [31].
3. Limited physical space. High land costs and limited physical space in DC have resulted in an emphasis on infill development or rehabilitation [32].
4. Mismatch between housing stock and incomes. From 2005 to 2012, the availability of luxury rental units increased and availability of affordable rental units decreased. Meanwhile, housing costs increased faster than incomes in the DC region [33] (Figure 4). DC's Gini coefficient as of 2016 was higher than that of all 50 states [34].

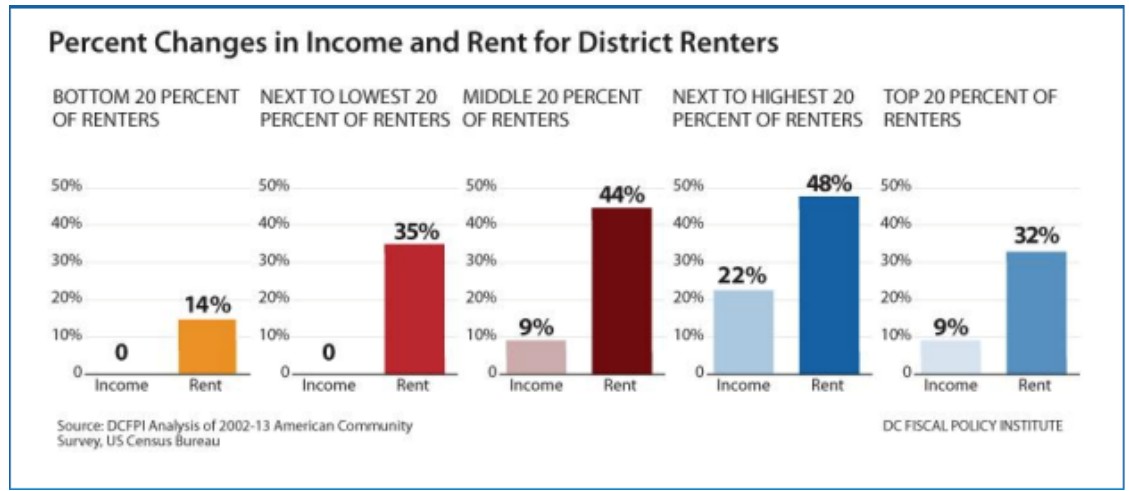

**Figure 4.** In Washington, DC, housing costs have outpaced household income from 2002 to 2013. Source: [35].

DC's Comprehensive Plan and the DC government are attempting to address the affordable housing shortage using a combination of regulations, subsidies, and loans at the district and federal level. These include inclusionary zoning (2009), requirement of community benefits in exchange for granting developers greater density or zoning variation under Planned Unit Developments (1958/2016), and establishment of the Tenant Opportunity to Purchase Act (1980) and District Opportunity to Purchase Act (2008). Subsidies and loans include the Housing Production Trust Fund (1988), which provides loans and grants to for-profit and non-profit developers to build or preserve existing affordable housing for targeted populations [36]; the DC Local Rent Supplementary Program (2007) for those with incomes below 30% of the area median income, or USD $32,600 for a family of four [37]; and property tax relief (for homeowners and renters) and deferral [38].

Some of these, like Tenant Opportunity to Purchase Act, are geared toward tenant empowerment in land transactions. Others are oriented toward increasing overall affordable housing stock. Critiques of the government's achievements include units created through inclusionary zoning are not targeted toward the lowest 20% of earners [39] and that some landlords are manipulating the Tenant Opportunity to Purchase Act system to the detriment of occupants [40]. Despite the policy measures in place, posters distributed around DC in advance of the June 2018 primary elections reflected the heated public discourse and implied a sense of anxiety around housing in Washington that warrants further study (Figure 5).

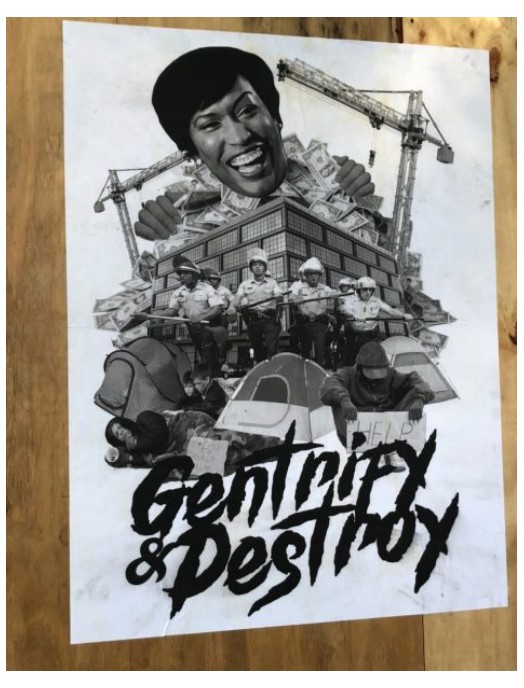
**(a)**

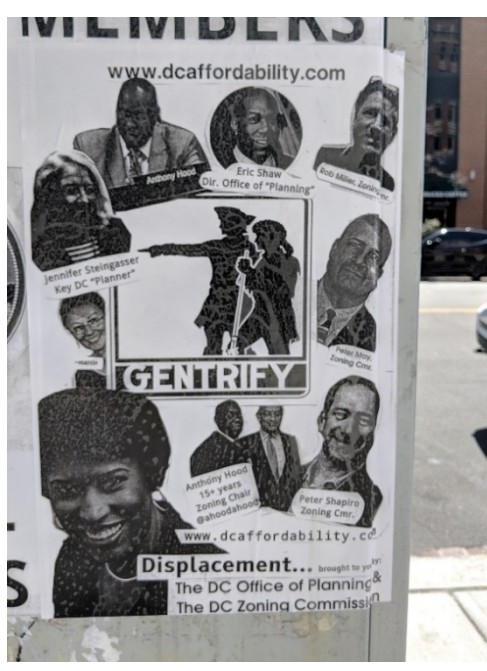
**(b)**

**Figure 5.** Posters distributed around DC in advance of the June 2018 primary elections reflect the heated public discourse around housing in Washington. (**a**) Anonymous wheatpaste flyer critiquing DC mayor. Source: [41]; (**b**) Anonymous wheatpaste flyer critiquing DC planning and zoning officials.

### 3.3.3. Potential Drivers of Responsive Displacement

As of late 2018, the DC Comprehensive plan is under revision again. Criticism of the plan includes weak commitments to affordable housing and a revised appeals process "that it makes it much harder for residents to appeal development they believe could worsen gentrification" [42]. Indeed, despite Zuk et al.'s caution to maintain an analytical distinction between the two, gentrification and displacement, have been conflated in the discourse around gentrification in DC, including through claims that the government is too friendly to developers targeting luxury rental markets instead of

addressing the housing needs of longer-term residents and larger families. A 2015 *Washington Post* poll on redevelopment found "a pervasive sense of economic anxiety among African Americans and those with annual incomes below USD $50,000" [43]. Alongside the economic anxiety are markers of Zuk et al.'s responsive displacement. One respondent of the *Washington Post* poll said new development has drained the city of some of its "homegrown feel and made it into an anonymous suburb" [43]. Hyra's 2015 study [7] of the Washington's Shaw/U Street neighborhood in DC's northwest quadrant echoed this sentiment: while redevelopment "has been associated with less crime, greater aggregate community income, higher property values, and increased social diversity", it also has brought "social costs for low-income residents: political and cultural displacement and feelings of community loss" (p. 1767). However, the last decade of scholarly literature that has addressed these socialized perceptions could be expanded to provide a methodological framework for accurately evaluating the correlation between gentrification and displacement, including responsive displacement, as housing pressures in select U.S. cities continue to mount and cities form policy responses.

*3.4. Assessing Mobilities in DC through the Lens of the Southwest Quadrant*

3.4.1. Southwest: What Does Renewal of Urban Renewal Mean for Residents?

Like Washington as a whole, the SW quadrant is again going through a period of change, fueled by a hybrid of public and private investment. The shifting composition of SW DC reflects DC history. After 50 years of population loss, in the early 2000s, DC started to gain population again. The population increase included both African-Americans, Caucasians, and others, although for the first time in decades, by 2015, DC was no longer a majority African-American city (Table 1).

**Table 1.** Composition of Southwest DC (urban renewal zone) from 1950 to 2000.

|  | 1950 | 1970 | 2000 |
| --- | --- | --- | --- |
| Population | 21,050 | 9,427 | 8,802 |
| Black | 69% | 32% | 57% |
| White | 31% | 67% | 33% |
| Median household income (in 1999 dollars) | $14,000 | $53,000 | $43,000 |
| Owner-occupied housing units | 15% | 22% | 35% |
| Renter-occupied housing units | 81% | 76% | 59% |
| Mean gross rent (in 1999 dollars, monthly) | $239 | $938 | $721 |

Note: Based on U.S. census data for the tracts that most closely align with the residential portion of the urban renewal zone: in 1970, this was census tracts 60.1, 61, and 63.01. Source: Excerpted from [21].

The composition of the Southwest mirrored the shifting composition of Washington, DC from 1950 to 2000, including a population decline from 1950 to 2000, flips in majority/minority population, and decline in property values and household income from 1970 to 2000. The patterns of economic and racial segregation in DC demonstrated in Figure 3 are largely replicated in SW, with higher income residents clustered along the waterfront and a higher concentration of African-Americans and lower income residents adjacent to the east, where public housing is located (Figure 6).

Housing (rent and purchase) costs in SW were slower to rise compared to the DC average in the early part of the 21st century. Factors included lack of commercial services, an informal reputation as a senior village for its older than average population, and the relative isolation of the neighborhoods from the rest of DC as a result of the 395 freeway—circumstances inherited from the urban renewal plan. The SW is now catching up in terms of population and cost of living with other areas of the city, as are the financial circumstances of SW's residents. Medium income in 2017 was USD $89,792, approximately USD $60,751 in 1999 dollars [44]. In 2010, census tracts 60.1, 61, and 63.01 were approximately 47% black and 40% white, with an increasing percentage of Hispanic and Asian/Pacific Islanders [40]. Given the neighborhood changes, it is well placed for a more in-depth study of the incidence of displacement and potential causes.

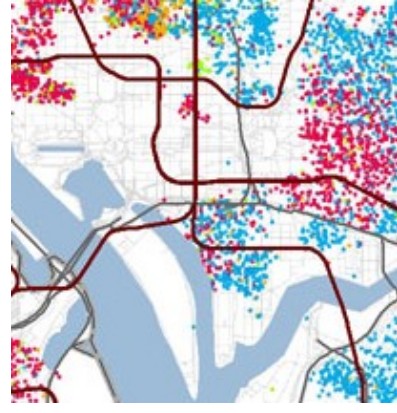

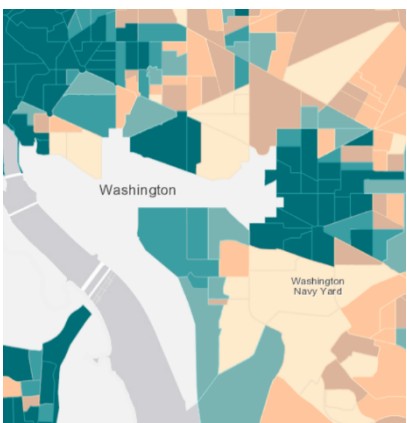

Key: ● = white; ● = black; ● = Asian; ● = Hispanic

Key: ● $24,000 or less; ● $24,001 - $39,000; ● $39,001 - $53,000; ● $53,001 - $68,000; ● $68,001 - $82,000 ● More than $82,000

Key: ● Tract gentrified; ● Tract did not gentrify; ● Tract not eligible to gentrify

**Figure 6.** Detail from Figure 3, SW quadrant.

In SW, uncertainty over change has manifested through opposition to developments both from homeowners, protecting their property values who value lower-density cities, and long-time residents who fear being pushed out of their neighborhoods if development raises housing costs, particularly for renters and fixed income owners [33]. Community engagement efforts undertaken by city officials in the development of the SW small area plan included community meetings, an advisory committee, focus groups, consultation with other District agencies, input from neighborhood groups, an extensive review of existing plans, and a site and market analysis [45]. The Plan states: "Southwest will remain an exemplary model of equity and inclusion—a welcoming and engaged community that celebrates and retains a mix of races, ages, and income levels and enhances well-being for all amidst neighborhood growth and change" [4] (p. 5). Yet, as shown in Figure 5, the messages in the public space evince a physical jockeying for space, indicating that the underlying power struggles that marked SW's past may remain.

### 3.4.2. Case Studies in SW

Two potential case studies in SW are presented here for interested researchers, with the positing that they would serve as case studies to elucidate the economic and socio-cultural factors shaping the actions of present and prospective residents and decision-makers and residents' mobilities. They are described in brief here.

Wharf Development

The Wharf development is a USD $2 billion-dollar public-private partnership. The District government provided close to USD $300 million in subsidies and expenditures to support the Wharf redevelopment on underutilized land near the SW waterfront, adjacent to the country's longest-running municipal fish market. The developers obtained a free 99-year lease of city-owned land on 27 acres that was valued at $95 million at the time of disposition. Although the first phase had 131 affordable units out of 649 units, the District lowered overall affordability requirements during negotiations from 30% of total units to, by one count, just over 10%, an echo of the past negotiations the RLA undertook with developers in the 1960s. Additional units have been earmarked for workforce housing bracketed for higher earning families—100% and 120% of annual median income (AMI) instead of the 30% and 60% AMI thresholds of the affordable units [46]. Additionally, it is perhaps ironic that one of the letters submitted for consideration during the public hearings on the development came from Tiber Island—the condominium complex built as part of the 1960's renewal project. Residents of Tiber Island and the nearby Gangplank Marina liveaboard community have raised concerns about the Wharf's impending Phase 2 including the pricing out of families and disruption to neighborhood character [47].

Greenleaf Gardens

Greenleaf was built in 1959 by the DC Housing Authority and currently contains 493 public housing units spread over 23 buildings. It is a 15-acre public housing project that straddles M Street, SW [44]. The DC Housing Authority built Greenleaf adjacent to the original SW Urban Renewal Zone to secure affordable housing for displaced residents [1]. It is adjacent to several plots that are currently in the process of in-fill development on the site of the old Waterfront Mall, which was built in the 1970s and demolished in 2007.

The SW Neighborhood Plan highlights the Greenleaf redevelopment as an opportunity within its broader plan to build a "model community" that "remains balanced: growing families, retaining public housing, supporting affordability, and improving the accessibility of infrastructure" [4] (71). A Request for Qualifications (RFQ) released by the DC government in December 2017 cited physical obsolescence and high maintenance costs as the rationale for redevelopment [48]. The RFQ outlines a "build first", zero displacement model to avoid displacement of Greenleaf residents. The RFQ's envisioned mix of market-rate, affordable, and possibly workforce housing included a stipulation that all 493 existing units be replaced at the same level of affordability. In community meetings, residents and neighbors have voiced concerns about unit size, space for community services, and eligibility requirements. As of the time of this writing, the DC government had not yet issued a formal Request for Proposal [49].

## 4. Conclusions

When the Anacostia Community Museum selected "Right to the City" for its exhibit title, it was taken in honor of "ordinary Washingtonians [who] have helped shape and reshape their neighborhoods in extraordinary ways" [50]. However, it may also be seen as aspirational, an aspiration held by the other global cities such as Mexico City and Montreal who have adopted tenets of a "right to the city" philosophy based on "solidarity, freedom, dignity, equity, and social justice" [51]. Despite affordable housing policies, Washington, DC faces considerable challenges if it wishes to empower its citizens to enact the Right to the City tenets. These challenges include a high demand for real estate in the city, a rapidly challenge demographic that appears to be exacerbating social stressors, and a contentious understanding of who embodies the Washington citizen [7,15,31–33]. It is hypothesized that the language of urban change is not adequately nuanced to capture the many factors at play—factors not explicit in euphemistic terms like revitalization, renewal, and gentrification. Despite historical lessons, Washington decision-makers have not yet created the mechanisms to ensure all citizens have the rhetorical and physical space to assert their presence and plan jointly for change. Among

identified analytical gaps are the understanding of the relationship between the historical dimension of place-making and the impacts of redevelopment, an examination of mobilities on a smaller scale, and the trade-offs between social harmony goals, economic growth, and/or diversification, and a methodological framework for accurately evaluating the correlation between gentrification and displacement that also considers incidence of responsive displacement.

As Sturtevant noted, it is extremely complex to analyze the redistribution of populations within a metropolitan area based on available census data [12]. Consequently, this article stops short of forecasting the type of mobilities that the present-day redevelopment plans for SW may trigger. However, the literature suggests this may occur, and posters like "Gentrify and Destroy" (Figure 5a) point to a long historical memory and latent anxieties that indicate Washington, DC is at another turning point in a continuous history of redevelopment and renewal.

**Funding:** This research received no external funding.

**Acknowledgments:** The author thanks Maeve Quigley, Rik Hartog, and Nathan Brown for review of the original conference presentation.

**Conflicts of Interest:** The authors declare no conflict of interest. The funders had no role in the design of the study; in the collection, analyses, or interpretation of data; in the writing of the manuscript, or in the decision to publish the results.

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
