# Peer review of "One City for All? The Characteristics of Residential Displacement in Southwest Washington, DC"

_land, doi:10.3390/land8020034_

Round 1

Reviewer 1 Report

This research tackles a very important topic about inclusive cities and its urban governance, but in its current format, the research paper stands as a subjective viewpoint with almost no theoretical framework to guide methodological investigations. The researcher seems overwhelmed by proposing potential future research rather than conducting rigorous research with a well-defined theoretical framework. So, it needs radical restructuring to phrase a clear problem with a justified method and investigation tools within a concrete and focused theoretical framework that tackles, in depth, the investigated topic(s) being gentrification, displacement and/or governance and their relationship with the urban sustainability debate. More comments are as follows:

-   The author would benefit from allocating a subsection for the research problem, questions, and objectives. The reader actually cannot easily find out the main questions or objectives of the research as they are scattered in more than one spot, each time with different phrasing. For example, in line 66: “this article encourages scholars to consider the ongoing negotiations for access to SW through several high-profile redevelopment projects as fertile grounds for in-depth community-level study” other phrases in section 2 might stand for the research objectives. Also, the introductory statement in Section 3 is about the purpose of the research, and so on.

-   Section 2 is inadequately short. More elaboration is needed about the adopted case study method and its tools for the analysis such as statistics, maps and so on. Also, the adopted method is not justified. For example, why interviewing stakeholders was not used for collecting primary data instead of relying only on secondary data …etc.

-   In Section 3, the author mentioned Land Governance. This is another issue that has not been introduced in the literature review section in the Introduction. Is the focus on gentrification or displacement? as the author claimed they are different in the introduction section. Or it is (or should be) on governance? This needs to be more elaborated in the whole manuscript.

-   Gentrification appears in section 3.3.1 with weak theoretical linkage with the displacement or governance discourse. Line 291, shows the two phenomena of displacement and generation are conflated, so, how the author deals with that, and where is the governance in the debate?

-   It seems that the SW is actually moving from the harmoniously populated zone (mainly Black) into a more socially-mixed one. This, from a social sustainability point of view, might be a positive outcome!. This needs to be discussed especially with the casual mention of urban sustainability in the research. The title of the paper: One city for all?; sounds biased towards anti-social mix strategies despite its success as mentioned by the author in line 299 “Hyra’s 2015 study of Washington’s Shaw/U Street neighborhood in the DC’s northwest quadrant … redevelopment “has been associated with less crime, greater aggregate community income, higher property values, and increased social diversity”, even with some social harmony cost. This is affirmed by information in Table 1 illustrating that the “population increase included both African-Americans, Caucasians, and others, although for the first time in decades in 2015 DC was no longer a majority African-American city. This has been affirmed by the Plan statement in line 335. So, apparently, the redevelopment Plan sounds good from a social diversity and mix point of view!

-   The tone of the research on some occasions becomes very subjective! One example is what the author stated in line 337 “Yet, as shown in Figure 7, the messages in the public space evince a physical jockeying for space that indicates the underlying power struggles that marked SW’s past may remain.” What is the percentage of those who oppose the redevelopment plan? What about those who support it?

-   The location of the [3.4.2. Case studies in SW] is not suitable as they should be an inherent part of the study not only...warrant future investigation!

-   In various locations in the research, the author indicated many issues that “... warrant further study”. Also, the sentence in line 326 and Section 3.4.2. Actually, it is better if the Further Study came as a concluding paragraph in the Conclusion section while focusing on the research topic and its findings in other sections.

-   In the Conclusion section. the author mentioned that: “and that Washington decision-makers create the mechanisms to ensure all citizens have the rhetorical and physical space to assert their presence and plan jointly for change”, but how can this happen? what are the suggested mechanisms for the “assertion for presence and joint planning for change”? This needs to be included in the debate and might be the core of the investigations.

-   Some claims in the Conclusion have not been proven in the text, at least in a rigorous and non-biased manner. For example, the author claims: “Despite affordable housing policies, Washington, DC faces considerable challenges if it wishes  to empower its citizens to enact Right to the City tenets; these challenges include a high demand for real estate in the city (where is the evidence for this claim in the text?), and a rapidly challenge demographic that appears to be exacerbating social stressors and a contentious understanding of who embodies the Washington citizen (Again, where is the concrete evidence for this claim in the text?) a piece of a local Newspaper is not enough as a rigorous evidence!

-   All in all, the reader is ultimately left with no answer for an ambiguous question in the title (is it one city for all?)   

-   It is better to have color legends for Figures 3a,b,c. The caption is too long!

-   Figure 5 can be removed. It is caption is distracting with a marginal question!

-   The caption of Figure 6 needs more elaboration.

-   Figure 7 can be removed.

-   In line 123, Areas B, Area C-1, and Area C should be shown on the map in figure 1 or another relevant figure.

-   in line 50, remove the repeated “that”; Research by Chapple (2017) shows that that...

Author Response

Thank you for your thoughtful and detailed comments. Please find below responses, with corresponding edits made in the text.

This research tackles a very important topic about inclusive cities and its urban governance, but in its current format, the research paper stands as a subjective viewpoint with almost no theoretical framework to guide methodological investigations. The researcher seems overwhelmed by proposing potential future research rather than conducting rigorous research with a well-defined theoretical framework. So, it needs radical restructuring to phrase a clear problem with a justified method and investigation tools within a concrete and focused theoretical framework that tackles, in depth, the investigated topic(s) being gentrification, displacement and/or governance and their relationship with the urban sustainability debate.

Response 1: The author recognizes the methodological limitations of this paper. This paper is an elaboration of a presentation given in the panel session “Land Governance in the Global North: Pointing the Lens at the Developed World” at the 2018 LANDAC Conference 2018: “Land Governance and (Im)mobility: Exploring the nexus between land acquisition, displacement and migration. The panel sought papers that “examine[d] changes in land governance in the Global North”. The author does not feel that “overwhelmed” is a fair assessment as it was precisely the intent to introduce a basic historical narrative to a broader array of practitioners who may have the quantitative tools and qualitative framework to conduct the more rigorous analysis sought by the Reviewer.

The author concurs with the Reviewer that there is great possibility for a paper that tackles in depth the topics listed above; however in the time granted for revision and the resources at hand it is not possible to, essentially conduct a new investigation to be captured here.

The author would benefit from allocating a subsection for the research problem, questions, and objectives. The reader actually cannot easily find out the main questions or objectives of the research as they are scattered in more than one spot, each time with different phrasing. For example, in line 66: “this article encourages scholars to consider the ongoing negotiations for access to SW through several high-profile redevelopment projects as fertile grounds for in-depth community-level study” other phrases in section 2 might stand for the research objectives. Also, the introductory statement in Section 3 is about the purpose of the research, and so on.

Response 2: A new introduction has been added to section 2, and the last sentences of Section 1 revised to match. The introductory statement in 3 was redundant with Section 2 content and has been eliminated.

Section 2 is inadequately short. More elaboration is needed about the adopted case study method and its tools for the analysis such as statistics, maps and so on. Also, the adopted method is not justified. For example, why interviewing stakeholders was not used for collecting primary data instead of relying only on secondary data …etc.

Response 3. Clarification of scope and intent has been added to the methodology section.

In Section 3, the author mentioned Land Governance. This is another issue that has not been introduced in the literature review section in the Introduction. Is the focus on gentrification or displacement? as the author claimed they are different in the introduction section. Or it is (or should be) on governance? This needs to be more elaborated in the whole manuscript.

Response 4. The author mentioned land governance, phrased as such, only inasmuch as the content of this paper was originally presented at a conference entitled “Land Governance and (Im)mobility: Exploring the nexus between land acquisition, displacement and migration.” References to planning and decision-making have been sharpened throughout.

Gentrification appears in section 3.3.1 with weak theoretical linkage with the displacement or governance discourse. Line 291, shows the two phenomena of displacement and generation are conflated, so, how the author deals with that, and where is the governance in the debate.

Response: The role of governance (through policy decision-making) is introduced in lines 273-290, and additional detail has been added to lines 314-316.

It seems that the SW is actually moving from the harmoniously populated zone (mainly Black) into a more socially-mixed one. This, from a social sustainability point of view, might be a positive outcome!. This needs to be discussed especially with the casual mention of urban sustainability in the research. The title of the paper: One city for all?; sounds biased towards anti-social mix strategies despite its success as mentioned by the author in line 299 “Hyra’s 2015 study of Washington’s Shaw/U Street neighborhood in the DC’s northwest quadrant … redevelopment “has been associated with less crime, greater aggregate community income, higher property values, and increased social diversity”, even with some social harmony cost. This is affirmed by information in Table 1 illustrating that the “population increase included both African-Americans, Caucasians, and others, although for the first time in decades in 2015 DC was no longer a majority African-American city. This has been affirmed by the Plan statement in line 335. So, apparently, the redevelopment Plan sounds good from a social diversity and mix point of view!

Response 6: The author is not intending to dismiss out of hand the positive consequences of urban change, some of which are noted in lines 43 to 66. However, the author questions the Reviewer’s reading of the Hyra and his/her assertion that social harmony cost is of neglible bearing on the evaluation of success of social-mix strategies. Indirectly the Reviewer raises an interesting line of inquiry – valuation of long and short term impacts of social harmony cost; the author is not aware if this has been attempted.

The author does not understand the correlation between population change (historical) and the value of the Redevelopment Plan (prospective) that the Reviewer is making, as the Redevelopment Plan was not a driver of the city-wide demographic change from 2000 to 2015 – it was finalized in 2017. The author notes that the initial SW urban renewal did result in more mechanisms to facilitate social mixing, including some of the first racially integrated housing complexes in the city; however, economic pressures rendered legally feasible mixing to be economically infeasible for former residents of the area who moved and were effectively “ghettoized” into public housing (177 to 184). This is, the author understands, one of the underlying critiques of gentrification. The author concurs that on paper the Redevelopment Plan for Greenleaf Gardens sets out to rectify the imbalance between economic accessibility and social desirability. But given the history (1960’s and with the Wharf) in the area of affordable housing provisions that would have secured social-mix opportunities being reduced, the question mark in the city seems warranted as it points to the gap between the tone of government-produced policy and the actual distribution of housing. The author notes that the Reviewer seems to desire the removal of references to “subjective” critique, but doing so perpetuates the marginalization of some indirect drivers of displacement enumerated by Zuk  

The tone of the research on some occasions becomes very subjective! One example is what the author stated in line 337 “Yet, as shown in Figure 7, the messages in the public space evince a physical jockeying for space that indicates the underlying power struggles that marked SW’s past may remain.” What is the percentage of those who oppose the redevelopment plan? What about those who support it?

Response 7: The author did not have the means to conduct a survey or otherwise collect the data requested by the reviewer. However, the author disagrees with the assertion that examination of physical evidence has no place in the formation of a hypothesis.

The location of the [3.4.2. Case studies in SW] is not suitable as they should be an inherent part of the study not only...warrant future investigation!

      Response: 8: Detailed case study analysis is recommended for interested researchers but field-based work was outside of the scope of this paper. The introduction to 3.4.2 has been revised.

In various locations in the research, the author indicated many issues that “... warrant further study”. Also, the sentence in line 326 and Section 3.4.2. Actually, it is better if the Further Study came as a concluding paragraph in the Conclusion section while focusing on the research topic and its findings in other sections.

Response 9: Identified recommended areas for further study have been consolidated or otherwise reiterated in the Conclusion.

In the Conclusion section. the author mentioned that: “and that Washington decision-makers create the mechanisms to ensure all citizens have the rhetorical and physical space to assert their presence and plan jointly for change”, but how can this happen? what are the suggested mechanisms for the “assertion for presence and joint planning for change”? This needs to be included in the debate and might be the core of the investigations.

Response 10: The author concurs that this is a fertile subject for investigation, but as the author is not a political scientist, concrete investigation of participatory planning methodologies was outside the scope of this inquiry.

Some claims in the Conclusion have not been proven in the text, at least in a rigorous and non-biased manner. For example, the author claims: “Despite affordable housing policies, Washington, DC faces considerable challenges if it wishes  to empower its citizens to enact Right to the City tenets; these challenges include a high demand for real estate in the city (where is the evidence for this claim in the text?), and a rapidly challenge demographic that appears to be exacerbating social stressors and a contentious understanding of who embodies the Washington citizen (Again, where is the concrete evidence for this claim in the text?) a piece of a local Newspaper is not enough as a rigorous evidence!

Response 11: Evidence for the high demand for real estate was provided in section 3.3.2. Interested readers may find additional supporting documentation in the literature cited in this section. Evidence as to the social stressors is found in Hyra (2015) and discussed in 3.3.3. These citations have been inserted here.  

All in all, the reader is ultimately left with no answer for an ambiguous question in the title (is it one city for all?) 

Response 12: As discussed above, the scope as defined did not seek to provide that answer, but simply to introduce the Lefebvrian discourse into the Washington context. It is an allusion to the Pledge of Allegiance of the United States (“one nation, with liberty and justice for all.”). One could point to historical evidence (promises versus actions taken during the SW Urban Renewal in the 1960s and again in the Wharf Planning stage) to make certain suppositions about history being destined to repeat itself, but that the author has not, in deference to the request for “non-biased evidence.”

It is better to have color legends for Figures 3a,b,c. The caption is too long!

Response 13: Color legends have been included here and in Figure 6. Note that length of caption is primarily due to Figure citation.

Figure 5 can be removed. It is caption is distracting with a marginal question!

Response 14: The “marginal question” is part of the title of the cited source for 5(a), it was not posed by the author. To maintain an accurate citation, it has not been removed. See response

As for Figure 5 itself, it is noted that Reviewer #2 has a similar observation regarding the value of Figure 5. However, the author would like to contend that both Figure 5 and 7, and in particular Figure 5, should be entered as documentation of the lived environment and manifestation of the “resentment and alienation” noted elsewhere in the city by Hyra (2015) and Green et. al. (2017). The author concurs with both Reviewers that such images do not stand on their own, and must be accompanied by source and contextual analysis. However, in particular given Reviewer #1’s hypothesis that “the redevelopment Plan sounds good from a social diversity and mix point of view”, it is contended that documentation of perceived cost in terms of social harmony must be entered into the equation, as it serves to problematize official narratives of continual improvement (for example, the Southwest BID’s planning documents).

The caption of Figure 6 needs more elaboration.

Response 15: Title and caption of Figure 6 have been revised and clarified.

Figure 7 can be removed.

Response 16: Removed, however please note response to comment 15 above.

In line 123, Areas B, Area C-1, and Area C should be shown on the map in figure 1 or another relevant figure.

Response 17: Additional archival research would be required to obtain precise coordinates of the three Areas; available maps with the areas demarcated were low quality. Due in part to the government shut-down and closure of relevant offices, it was not possible to obtain this information in the review period.

in line 50, remove the repeated “that”; Research by Chapple (2017) shows that that...

Response 18: Thank you, this has been corrected.

Reviewer 2 Report

Author presented a well written paper, that has a correct structure and a very well explained sequence of: “problem setting” - “state if the art” – “methodology” – “conclusions”.

The issue about gentrification in urban transformation is actual and it is a world wide problem. Unfortunately the paper is strictly connected to Washington context, both in problem description than in methodology. In other words, it could be an interesting “case study”.

This is the main flaw of the paper, and it effects the scientific soundness and the possibility to replicate the methodology in other contexts.

By the way, the presented example is surely interesting.

A minor consideration: figures 5 and 7 sound like “non technical”, and I do not fell the necessity to insert them in the text.

Author Response

Thank you for your comments. Please find below responses, with corresponding edits made in the text.

1. Author presented a well written paper, that has a correct structure and a very well explained sequence of: “problem setting” - “state if the art” – “methodology” – “conclusions”.

Response 1: Thank you.

2. The issue about gentrification in urban transformation is actual and it is a world wide problem. Unfortunately the paper is strictly connected to Washington context, both in problem description than in methodology. In other words, it could be an interesting “case study”. This is the main flaw of the paper, and it effects the scientific soundness and the possibility to replicate the methodology in other contexts.

Response 2: This paper is an elaboration of a presentation given in the panel session “Land Governance in the Global North: Pointing the Lens at the Developed World” at the 2018 LANDAC Conference 2018: “Land Governance and (Im)mobility: Exploring the nexus between land acquisition, displacement and migration. The panel sought papers that “examine changes in land governance in the Global North”. As the author’s primary discpline is history, it was outside of the scope of the paper to introduce a replicable methodology. Rather, the intent was to introduce the possibility of Washington case studies to a broader array of practitioners who may have the quantitative tools and qualitative framework to conduct the more rigorous analysis sought by the Reviewer.

Clarification of scope and intent has been added to the methodology section.

3. A minor consideration: figures 5 and 7 sound like “non technical”, and I do not fell the necessity to insert them in the text.

Response 3. It is noted that Reviewer #1 has a similar observation. However, the author would like to contend that both Figures, and in particular Figure 5, should be entered as documentation of the lived environment and manifestation of the “resentment and alienation” noted elsewhere in the city by Hyra (2015) and Green et. al. (2017). The author concurs with both Reviewers that such images do not stand on their own, and must be accompanied by source and contextual analysis. However, in particular given Reviewer #1’s contention that “the redevelopment Plan sounds good from a social diversity and mix point of view”, it is contended that documentation of perceived cost in terms of social harmony must be entered into the equation, as it serves to problematize official narratives of continual improvement (for example, the Southwest BID’s planning documents). In response to both sets of comments, Figure 7 has been removed.

Round 2

Reviewer 1 Report

The clarification made by the author in the revised version of the text has obviously clarified the expectation and the scope of the paper. The objectives of the research now are clearer. I still feel some subjectivity (maybe due to the lack of primary sources data) and the author might highlight/admit that in a Research Limitation sentence. But anyway, the research addresses a very interesting topic and I urge the author to consider it as an introductory effort for more rigorous in-depth investigations.